physical chemistry/materials science/energy

thermoelectric, bismuth telluride, mechanical alloying, spark plasma sintering

**Author for correspondence:**
Dennis Groeneveld
e-mail: dennis.groeneveld@imtek.uni-freiburg.de

This article has been edited by the Royal Society of Chemistry, including the commissioning, peer review process and editorial aspects up to the point of acceptance.

This paper is dedicated to the memory of Uwe Kruck.

# Time-dependent investigation of a mechanochemical synthesis of bismuth telluride-based materials and their structural and thermoelectric properties

Dennis Groeneveld[1], Jan D. Koenig[1,2], Michael Poschmann[3], Hendrik Groß[4], Wolfgang Bensch[5], Lorenz Kienle[4] and Jürgen Wöllenstein[1,2]

[1]Laboratory for Gas Sensors IMTEK - Department of Microsystems Engineering, University of Freiburg, Georges-Koehler-Allee 102, 79110 Freiburg, Germany
[2]Fraunhofer Institute for Physical Measurement Techniques IPM, Georges-Köhler-Allee 301, 79110 Freiburg, Germany
[3]Max-Planck-Institute for Chemical Energy Conversation, Stiftstrasse 34-36, 45470 Mülheim an der Ruhr, Germany
[4]Institute for Materials Science, Kiel University, Kaiserstrasse 2, 24143 Kiel, Germany
[5]Institute of Inorganic Chemistry, Kiel University, Max-Eyth-Strasse 2, 24118 Kiel, Germany

DG, 0000-0001-7532-5342

Here, we report on the time dependence of a synthesis procedure for generation of both n- and p-type bismuth telluride-based materials. To initiate the reaction, the starting materials were first mechanical pre-reacted. The Rietveld refinements of X-ray diffraction (XRD) data collected after different milling times demonstrate that $Bi_2Te_3$ was formed after only 10 min, and longer milling times do not alter the composition. To complete the phase formation, the powders were treated by field-assisted sintering and heat treatment afterwards. The effect of this fast procedure on the structural and thermoelectric properties was investigated. Samples were obtained with relative densities above 99%. A clear preferred orientation of the crystallites in the samples is evidenced by Rietveld refinements of XRD data. The thermoelectric characteristics demonstrate a good performance despite the short milling time. Further, it was demonstrated for this fast synthesis that the physical transport properties can be varied with well-known n- and p-type dopants like $CHI_3$ or Pb. For these non-optimized materials, a ZT value of 0.7 (n-type) and 0.9 (p-type) between 400 and

450 K was achieved. The long-term stability is demonstrated by repeated measurements up to 523 K showing no significant alteration of the thermoelectric performance.

# 1. Introduction

Thermoelectric materials are used in thermoelectric devices to directly convert thermal energy into electrical energy and vice-versa as thermoelectric generators or Peltier coolers, respectively. The conversion efficiency of these materials depends on the so-called thermoelectric figure of merit: $ZT = \sigma \cdot \alpha^2 \cdot \kappa_{\text{tot}}^{-1} \cdot T$ [1], with the electrical conductivity $\sigma$, the Seebeck coefficient $\alpha$, the thermal conductivity $\kappa_{\text{tot}}$ and the absolute temperature $T$. The thermal conductivity of solid-state materials consists mainly of the lattice thermal conductivity by phonons $\kappa_1$ and the thermal conductivity of charge carriers $\kappa_e$.

Bismuth telluride-based materials, their solid solutions or layered assemblies are the state-of-the-art thermoelectrics around room temperature [2]. The rhombohedral structure of $Bi_2Te_3$ (space group $R\bar{3}m$) can be described by a hexagonal unit cell in which layers of identical atoms are stacked in the sequence -$Te^1$-Bi-$Te^2$-Bi-$Te^1$- parallel to the crystallographic $c$-axis [3,4]. The unit cell consists of quintuple layers weakly bound by van der Waals interactions between $Te^1$-$Te^1$, leading to poor mechanical stability of single crystals. The $Te^1$-Bi bonds are covalent, while the Bi-$Te^2$ bond has an appreciable ionic character. If Te is substituted with Se, the Se replaces the $Te^2$ lattice sites, increasing the ionicity of these bonds [4,5]. The layered structure of $Bi_2Te_3$-based solid solutions results in a strong anisotropy of the thermoelectric properties. As a result, the electrical conductivity and the thermal conductivity perpendicular to the $c$-axis are about four and two times larger than parallel to the $c$-axis, respectively [3,6]. Since the Seebeck coefficient is nearly isotropic, the $ZT$ value is around two times larger perpendicular to the $c$-axis.

Many efforts have been made in the last decades to improve the thermoelectric performance, including nanoalloying [7–10], chemical vapour deposition [11–13], sputtering techniques [14,15], superlattice formation [16–20] or intermixing of different phases [21,22]. Because $Bi_2Te_3$, $Sb_2Te_3$ and $Bi_2Se_3$ are isostructural, formation of p-type $(Bi,Sb)_2Te_3$ and n-type $Bi_2(Te,Se)_3$ solid solutions is easily achieved resulting in a decrease of the lattice thermal conductivity [23–25]. The $ZT$ value can also be improved by optimizing the charge carrier concentration through targeted doping or substitution. In $Bi_2(Te,Se)_3$, halogen atoms such as I replace Te atoms and act as electron donors [26]. The same principle can be applied to the p-type material $(Bi,Sb)_2Te_3$. If Bi is substituted with an electron acceptor such as Pb, the hole concentration can be increased [27].

Usually, bismuth telluride-based alloys are prepared by energy- and time-intensive synthesis methods, e.g. zone melting [28,29]. Several powder metallurgical methods have been studied in the last few decades [30], in order to obtain outstanding thermoelectric materials using a fast and economic large-scale production. Thereby, the combination of mechanical alloying (MA) with hot pressing (HP) or field-assisted sintering technique (FAST)/spark plasma sintering (SPS) turned out to be a promising method [31–34]. MA is a simple, fast and easily scalable method that can be used to improve the thermoelectric properties [31,33,35]. FAST represents a densification technique to achieve high densities and optimized properties and is significantly faster than HP [36–38]. Another advantage of the combination of MA and FAST is that the materials produced are usually isotropic. Thereby, they have better mechanical properties than highly oriented samples, making the fabrication of thermoelectric devices easier [30,31]. Grain growth or the growth of (nano-)precipitates present extreme difficulties in consolidating bulk nanostructures via these powder synthesis routes in order to obtain high thermoelectric properties [39,40].

A fast reaction time at low temperatures can help to avoid grain as well as precipitation growth and therefore can lead to a synthesis of thermally stable nanostructured materials with high thermoelectric figure of merits ZT. Nevertheless, a detailed investigation of the time dependences of milling time as a pre-reaction followed by a heat treatment afterwards wasń't done before to our knowledge.

Here, we report the time-dependent investigation of the low-temperature synthesis of $(Bi,Sb)_2(Te,Se)_3$ prepared via a pre-reaction by MA followed by FAST/SPS to complete the phase formation. The influences of the chemical composition, substitution, annealing and texture on the physical transport properties were investigated.

# 2. Experimental

For the preparation of the solid solutions, commercial high-purity elemental powders of Bi (99.9999%, pieces), Te (99.999%, pieces), Se (99.999%, granules 2–4 mm), Sb (99.9999%, shots 1–3 mm) and Pb

(99.9+ %, shots 0.4–0.8 mm) from Chempur and $CHI_3$ (99%, powder) from Alfa Aesar were used as starting materials. Before milling, Bi, Te and Se were ground into a fine powder. According to the targeted nominal composition, the amounts of the reactants were placed in a tungsten carbide (WC) milling jar (250 ml) with WC milling balls in an argon-filled glove box ($O_2$ < 1 ppm, $H_2O$ less than 1 ppm). The ball diameter was 5 mm and the ball-to-powder weight ratio 10 : 1. The MA was performed in a planetary ball mill PM 100 from Retsch GmbH at a rotation speed of 450 rpm and a total milling time of 90 min, alternating milling for 1 min and pausing for 1 min. After milling, the powders were sintered into discs with a standard diameter of 20 mm and a final height of around 11–12 mm using a FAST equipment 'HP D 5' from the company FCT Systeme GmbH. The n-type materials were prepared under argon atmosphere at 723 K and a pressure of 22 MPa. At 753 K and a pressure of 51 MPa, the p-type materials were sintered. For both materials, a holding time of 30 min was chosen. After sintering, all samples were annealed for 24 h at 523 K under argon atmosphere. Finally, the samples were cut into rectangular pieces ($10 \times 10 \times 1.4$ mm) parallel (∥, called out-of-plane) and perpendicular (⊥, called in plane) to the FAST pressing direction. In total, three n-type and four p-type materials were prepared with compositions listed in table 1.

To track the effect of milling on the elements during synthesis of $Bi_{0.3}Sb_{1.7}Te_3$, after milling times of 1 min, 3 min, 5 min, 10 min and 30 min, samples were extracted and characterized with X-ray diffraction (XRD). Therefore, the obtained powder was mixed with amorphous $SiO_2$ in a weight ratio 1 : 2 to reduce the effect of X-ray absorption and filled in capillaries (glass, 0.2 mm diameter, wall thickness 0.01 mm, W. Müller). XRD patterns were collected with a PAN'alytical Empyrean MPD using Cu $K_\alpha$ irradiation. A focusing mirror, fixed 1/4° divergence slit, fixed 1/2° anti-scatter slit and 2.292° Soller slit was used to adjust the incident beam. Diffracted beams were altered by 2.292° Soller slit. Rietveld analyses of the powder patterns were done using the program Topas v6 [41] in combination with coding program jedit [42]. As initial structures data for Te, Sb, Bi and $Bi_2Te_3$ were used from ICSD 65692 [43], ICSD 9859 [44], ICSD 64703 [45] and ICSD 74348 [46].

XRD patterns of sintered samples were measured in reflection mode on a height-adjustable flat stage of a PAN'alytical X'Pert MPD using Cu $K_\alpha$ irradiation. To adjust the incident beam, a Goebel mirror and a set of fixed divergence slit 1/4°, fixed anti-scatter slit 1/2° and 2.292° Soller slit were used. The scattered beam was freed from diffuse X-rays by a parallel plate collimator. To measure XRD patterns of powdered samples, cuts of the pellets were ground and thoroughly mixed in a 1 : 2 weight ratio with amorphous $SiO_2$ to reduce influences of X-ray absorption on the diffraction patterns. The mixtures were filled into 0.2 mm capillaries (glass, W. Müller, wall thickness 0.01 mm) and the XRD patterns were collected with a PAN'alytical Empyrean MPD using Cu $K_\alpha$ irradiation. A focusing mirror, fixed 1/4° divergence slit, fixed 1/2° anti-scatter slit and 2.292° Soller slit was used to adjust the incident beam. Diffracted beams were altered by 2.292° Soller slit. Rietveld analyses of the powder patterns were done using the program Topas v. 6 [41] in combination with coding program jedit [42]. The instrument's profile function was determined by fitting the X-ray diffractogram of $LaB_6$ SRM 660c NIST standard using a Thomson-Cox-Hastings pseudo-Voigt profile function with an additional simple axial model as implemented in Topas. For the Rietveld refinements, crystallographic data of $Bi_{0.4}Sb_{1.6}Te_3$ published by M. M. Stasova *et al.* [47] and of $Bi_2Te_2Se$ published by S. Nakajima *et al.* [48] were used as initial values for refinements. Preferred orientation of crystallites within pelletized samples was evaluated via single fitting of (006) and (2$\bar{1}$0) reflections. The reflection intensities were compared to peak intensities of powdered samples, and the preferred orientation parameter was calculated using the March-Dollase approach [49,50] supplemented by E. Zolotoyabko [50].

The bulk densities of the samples were determined with an Archimedes set-up using ethanol at room temperature, with the resulting values being averaged from four measurements. The densities were measured of the whole samples before sawing. Relative densities were calculated as a ratio, comparing measured density and calculated density via Rietveld refinements $\rho_{obs}/\rho_{XRD}$. The charge carrier concentration and the mobility were determined at room temperature by Hall effect measurements using the van der Pauw method (uncertainty approx. equal to 10%). The electrical conductivity and the Seebeck coefficient were measured from 298 K to 523 K using the SBA 458 Nemesis device® under $N_2$ atmosphere. The thermal diffusivity was measured with the Netzsch LFA 457 MicroFlash® apparatus also under $N_2$ atmosphere. According to the relation $\lambda(T) = \rho(T) \cdot c_p(T) \cdot a(T)$, with the density $\rho$, the specific heat $c_p$ and the thermal diffusivity $a$, the thermal conductivity $\lambda$ can be calculated. The specific heat can be determined by using a reference specimen and is shown like the thermal diffusivity in the electronic supplementary material, figures S6 and S7. Before starting the measurement, the samples were coated with graphite to increase the emissivity and the absorbance.

**Table 1.** Structural characteristics of pellet cuttings and ground powders obtained from XRD patterns via peak fitting and Rietveld analysis. Calculated errors are given in parentheses as deviations from the last digit.

| sample | orientation to pressing direction | chemical formula | preferred orientation $\eta$ % | cell parameter | | cell volume | strain $\varepsilon_0$ |
|---|---|---|---|---|---|---|---|
| | | | | $a/b$ Å | $c$ Å | Å$^3$ | % |
| 1_n ∥ | parallel | $Bi_2Te_{2.7}Se_{0.3}$ | 13.3(4) | 4.36049(3) | 30.3584(4) | 499.90(1) | 0.074(1) |
| 1_n ⊥ | perpendicular | + 0.1 wt% $CHI_3$ | 24.9(2) | | | | |
| 2_n ∥ | parallel | $Bi_2Te_{2.4}Se_{0.6}$ | 18.7(1) | 4.33481(4) | 30.2052(4) | 491.53(1) | 0.066(1) |
| 2_n ⊥ | perpendicular | + 0.1 wt% $CHI_3$ | 7.8(0) | | | | |
| 3_n ∥ | parallel | $Bi_2Te_{1.5}Se_{1.5\ 3}$ | 12.5(2) | 4.25886(4) | 29.74788(5) | 467.28(1) | 0.05(1) |
| 3_n ⊥ | perpendicular | + 0.1 wt% CHI | 17.8(4) | | | | |
| 4_p ∥ | parallel | $Bi_{0.3}Sb_{1.7}Te_3$ | 20.3(2) | 4.28476(4) | 30.4721(5) | 484.49(1) | 0.081(1) |
| 4_p ⊥ | perpendicular | | 25.7(2) | | | | |
| 5_p ∥ | parallel | $Bi_{0.5}Sb_{1.5}Te_3$ | 23.1(3) | 4.29716(7) | 30.4877(9) | 487.55(2) | 0.081(2) |
| 5_p ⊥ | perpendicular | | 22.3(1) | | | | |
| 6_p ∥ | parallel | $Pb_{0.01}Bi_{0.29}Sb_{1.7}Te_3$ | 19.9(0) | 4.28570(6) | 30.4951(6) | 485.07(2) | 0.064(1) |
| 6_p ⊥ | perpendicular | | 16.8(0) | | | | |
| 7_p ∥ | parallel | $Pb_{0.01}Bi_{0.49}Sb_{1.5}Te_3$ | 22.1(3) | 4.29554(5) | 30.5017(6) | 487.63(2) | 0.078(1) |
| 7_p ⊥ | perpendicular | | 18.9(1) | | | | |

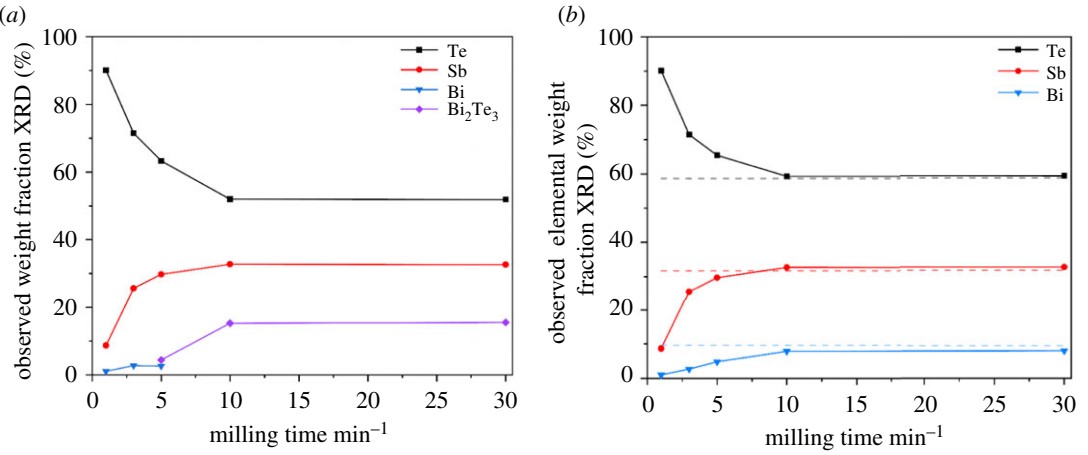

**Figure 1.** (a) Relative weight fractions of phases observed via Rietveld refinements in ex situ XRD experiments of a mixture of Te, Sb and Bi during the milling procedure. (b) Phase independent elemental weight fractions were observed via Rietveld refinements in ex situ XRD experiments of a mixture of Te, Sb and Bi during milling procedure. Expected weight fractions are marked as a dashed line in respective colour.

SEM surface and morphology analyses were performed in a Zeiss Gemini Ultra 55 Plus, equipped with an Oxford SD EDX detector. The absolute measurement uncertainty for reference-less EDX data can be expected at 1–2 at.%. Hardness measurements were performed at ambient conditions on a Leco V-100-C1 hardness tester using the Vickers method (ISO 6507) with a load of 5 kg (49.03 N) and a loading time of 15 s. The resulting values for hardness were averaged over eight data points per sample.

# 3. Results and discussion

## 3.1. Structural analysis

Applying ex situ XRD experiments on materials extracted from the milling process of the elements as the initial step of the synthesis of $(Bi, Sb)_2Te_3$, the phase composition can be analysed with Rietveld refinements (see electronic supplementary material, figures S1–S5 and table S1). During the milling process, a phase composition of $Bi_2Te_3$, Te and Sb is formed (figure 1) in the first 10 min. A longer milling time does not have any effect on the phase composition. Interestingly, antimony gains a relative weight fraction, while elemental Bi is almost not present. After a milling time of more than 5 min elemental, Bi is not observed in the material while reflections of $Bi_2Te_3$ occurred after 5 min or longer milling times. This observation evidences a fast reaction of Bi with Te under milling conditions while Sb stays inert. Another important fact is that with short milling times, the observed weight fraction of Te in XRD markedly exceeds the amount used for the synthesis. When determining the phase independent elemental weight fractions of Te, Sb and Bi (figure 2), it becomes clear that these relative amounts are observed due to inhomogeneity. Note, only a tiny portion of sample could be prepared in capillary XRD experiments and the small diameter of the capillary is selectively too small for bigger particles. Within the first 10 min, a homogenization of the material takes place; afterwards, a powdered sample with expected weight ratios is obtained. Summing up, within the first 10 min of milling, the mixture of Te, Bi and Sb homogenizes and forms a pre-reacted material consisting of Sb, Te and $Bi_2Te_3$.

After the sintering procedure, the pelletized samples were characterized as in-plane and out-of-plane cuttings via XRD. All the materials appear to be phase-pure $(Bi,Sb)_2(Te,Se)_3$ and polycrystalline, so the sintering procedure completes the pre-reaction during the milling procedure. Each sample exhibits a distinct preferred orientation of 8–26% of crystallites with a/b-plane in cutting direction (table 1).

This preferred orientation seems to be random and not associated with uniaxial pressing direction or elemental constituents. Rietveld refinements on powder XRD patterns collected of ground pellets show that all samples are phase-pure and crystalline with reflections in accordance with space group $R\bar{3}m$ (figure 4; electronic supplementary material, figures S8–S14). Increasing the contents of Bi and Te leads to an expansion of the unit cell parameters (table 1) [48,49,51]. The effect is less pronounced if

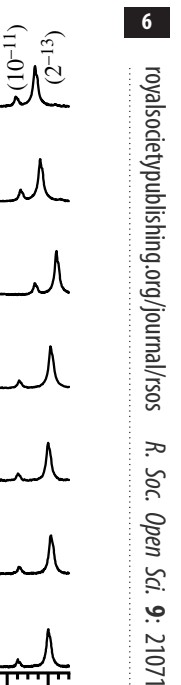

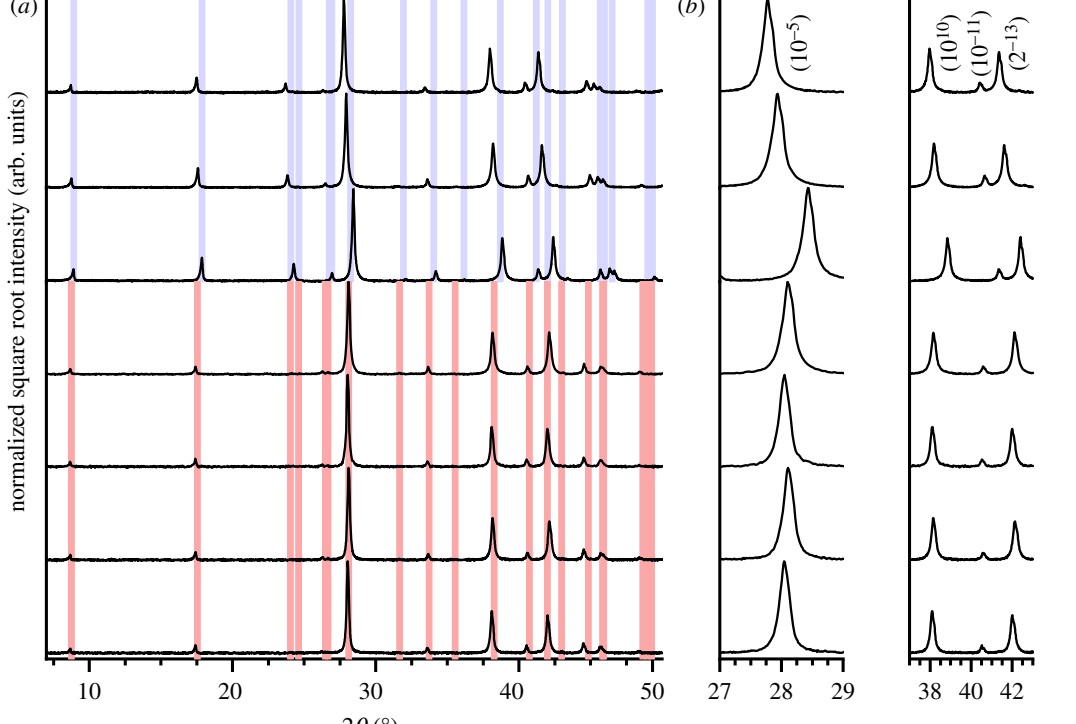

**Figure 2.** (*a*) XRD patterns observed of ground samples 1–7 after SPS top to bottom. Calculated reflection positions of $Bi_2Te_{1.5}Se_{1.5}$ [26] and $Sb_2Te_3$ [25] are marked as blue and red stripes, respectively. Background is subtracted for clarity. (*b*) Enlargement of the most prominent reflections with corresponding (hkl)-indices.

**Table 2.** SEM-EDX data and corresponding composition of the samples' surfaces after sintering. Both n-type materials match their theoretical compositions after sintering, while the p-type materials show minor deviations within the method's uncertainty.

| sample | chemical formula | element content in at.% | | | | | corresponding formula |
| --- | --- | --- | --- | --- | --- | --- | --- |
| | | Pb | Bi | Sb | Te | Se | |
| 1_n | $Bi_2Te_{2.7}Se_{0.3}$ + 0.1 wt% $CHI_3$ | — | 41(1) | — | 53(1) | 6(1) | $Bi_{2.09}Te_{2.7}Se_{0.31}$ |
| 2_n | $Bi_2Te_{2.4}Se_{0.6}$ + 0.1 wt% $CHI_3$ | — | 41(1) | — | 48(1) | 11(1) | $Bi_{2.05}Te_{2.7}Se_{0.55}$ |
| 4_p | $Bi_{0.3}Sb_{1.7}Te_3$ | — | 10(1) | 36(2) | 59(1) | — | $Bi_{0.51}Sb_{1.58}Te_3$ |
| 7_p | $Pb_{0.01}Bi_{0.49}Sb_{1.5}Te_3$ | 0(1) | 6(1) | 35(1) | 59(1) | — | $Pb_0Bi_{0.31}Sb_{1.78}Te_3$ |

Sb is substituted with Bi, because the difference in lattice parameters of $Sb_2Te_3$ [52] and $Bi_2Te_3$ [46] is small compared with the difference of lattice parameters for $Bi_2Te_3$ and $Bi_2Se_3$ [53].

In total, the unit cell volume increases as expected for substitution with heavier elements (table 1) [48,49,51,54]. Through substitution of Bi and Sb with Pb, the unit cell volume increases slightly. The isotropic microstrain is observed to be small in all samples but not correlated with elemental composition. The free parameter of the pnictogen position (0,0,z) increases slightly with higher Te content (see electronic supplementary material, table S2). As observed for the unit cell parameters, this effect is less pronounced when Sb is substituted by Bi and Pb. The chalcogen position (0,0,z) is only slightly affected with increased Se content (see electronic supplementary material, table S2) [48]. Substitution of Sb with Bi and Pb in the selected range has no pronounced effect on chalcogen positions.

A detailed SEM-EDX investigation of samples 1_n, 2_n, 4_p and 7_p before and after annealing revealed that all samples are free of foreign impurities and match their desired compositions within the uncertainty of the method (table 2). The samples' surfaces exhibited a visible grain structure, with particle sizes ranging from 1 to 2 µm for both n-type Bi-Te-Se materials and 2–4 µm for the p-type Bi-Te-Sb systems. Before annealing, all investigated samples appeared to be homogeneous, both in regard to structure and composition (figure 3; electronic supplementary material, figures S15–S18). For

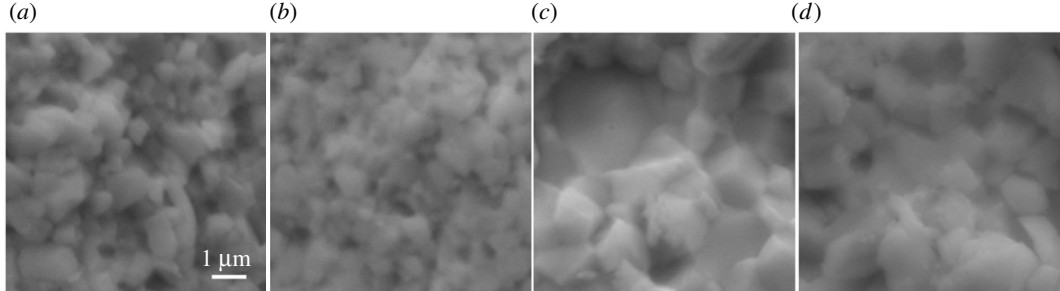

**Figure 3.** SEM surface micrographs of samples 1_n (*a*), 2_n (*b*), 4_p (*c*) and 7_p (*d*) before annealing, imaged at identical magnifications. All samples exhibit slightly grained surfaces, with the particle size increasing significantly from n-type to p-type materials.

the n-type materials, no significant changes could be observed after annealing. In the case of the p-type materials, however, deviations of 3–5 at.% from the ideal stoichiometry in their respective Bi content could be detected. While 4_p exhibited a slight excess of Bi in conjunction with a deficit in Sb before annealing, the situation appeared vice-versa for 7_p. For the latter case, the nominal lead content could not be detected via SEM-EDX, mainly due to its concentration being on the edge of the detection limit. Annealing of the p-type materials led to the elimination of the Bi-Sb deviations, but also oxidation of the samples' surfaces. In the case of 4_p, the oxygen appeared to have mostly been incorporated into the material's surface, while a few singular Sb-O crystallites could be found as well. The surface of the 7_p sample, on the other hand, was covered by crystallites with sizes ranging from 300 nm to 1 μm. EDX analyses showed a high content of O and Sb with the other constituents reduced to traces, leading to the assumption that oxidation had drawn out Sb from the bulk to form $Sb_2O_3$ crystallites. Apart from the surface oxidation, the elemental distribution of the p-type materials could be proven to be homogeneous as well.

## 3.2. Room temperature physical and thermoelectric data

The measured densities, the charge carrier concentrations and mobilities of the annealed samples are shown in table 3. It is obvious that the room temperature properties are almost independent from the orientation of the sample. The relative densities of over 97% demonstrate that highly densified samples can be produced with the presented synthesis method. By changing the amount of Te and Se, the carrier concentration and mobility can be adjusted. The carrier concentration increases, while the mobility decreases with higher Se content. With the substitution of Bi by Sb, the carrier mobility decreases. To increase the carrier concentration of the p-type material significantly, a small amount of Bi was substituted with Pb. However, the addition of Pb reduced the mobility of the charge carriers. The values of the carrier mobilities are similar regardless of the orientation to the addition to the thermoelectric properties; the mechanical stability is a critical factor for the manufacturing of thermoelectric modules. The Vickers hardness at room temperature of the in-plane samples is shown in table 3. The in-plane hardness could be determined as around 1 GPa for the n-type materials, while the hardness of the p-type samples appears lower, in the range of 0.78–0.83 GPa. Due to the random orientation of the crystallites, the values are generally significantly higher compared to samples synthesized by other methods. For example, $Bi_2Te_3$-based solid solutions prepared by gas atomization, hot extrusion or zone melting have a Vickers hardness in the range of 0.26–0.63 GPa [55–63].

## 3.3. Influence of annealing on physical and thermoelectric data

Figure 4*a,b* clearly shows the effect of annealing on the temperature-dependent physical transport properties of $Pb_{0.01}Bi_{0.49}Sb_{1.5}Te_3$. Before annealing, the electrical conductivity appeared lower during cooling. The reverse trend was observed for the Seebeck coefficient, which increased slightly after heating. Due to the annealing the charge carrier mobility increases, while the charge carrier concentration decreases to a greater extent (table 3; electronic supplementary material, table S3). Therefore, the overall electrical conductivity decreases and the Seebeck coefficient increases after annealing. Stable physical transport properties could only be achieved after the sample had been annealed. With regard to the use of the materials in a module, the long-term stability of the samples is of decisive importance. To demonstrate the long-term stability of the thermoelectric properties, the

**Table 3.** Overview of the properties at room temperature for the synthesized thermoelectric materials.

| sample | orientation to pressing direction | chemical formula | bulk density g cm$^{-3}$ | relative density % | carrier concentration cm$^{-3}$ | carrier mobility cm$^2$ V$^{-1}$ s$^{-1}$ | Vickers hardness HV GPa |
|---|---|---|---|---|---|---|---|
| 1_n ∥ | parallel | $Bi_2Te_{2.7}Se_{0.3}$ | 7.77 | 99.12 | $4.4 \times 10^{19}$ | 149 | |
| 1_n ⊥ | perpendicular | + 0.1 wt% $CHI_3$ | | | $4.8 \times 10^{19}$ | 147 | 0.98 |
| 2_n ∥ | parallel | $Bi_2Te_{2.4}Se_{0.6}$ | 7.76 | 99.24 | $4.9 \times 10^{19}$ | 118 | |
| 2_n ⊥ | perpendicular | + 0.1 wt% $CHI_3$ | | | $5.2 \times 10^{19}$ | 123 | 1.01 |
| 3_n ∥ | parallel | $Bi_2Te_{1.5}Se_{1.5}$ | 7.69 | 99.17 | $6.0 \times 10^{19}$ | 100 | |
| 3_n ⊥ | perpendicular | + 0.1 wt% $CHI_3$ | | | $5.9 \times 10^{19}$ | 113 | 1.00 |
| 4_p ∥ | parallel | $Bi_{0.3}Sb_{1.7}Te_3$ | 6.92 | 99.75 | $1.3 \times 10^{19}$ | 313 | |
| 4_p ⊥ | perpendicular | | | | $1.4 \times 10^{19}$ | 323 | 0.82 |
| 5_p ∥ | parallel | $Bi_{0.5}Sb_{1.5}Te_3$ | 6.83 | 99.79 | $6.9 \times 10^{18}$ | 348 | |
| 5_p ⊥ | perpendicular | | | | $7.4 \times 10^{18}$ | 342 | 0.83 |
| 6_p ∥ | parallel | $Pb_{0.01}Bi_{0.29}Sb_{1.7}Te_3$ | 6.92 | 99.87 | $7.5 \times 10^{19}$ | 227 | |
| 6_p ⊥ | perpendicular | | | | $7.1 \times 10^{19}$ | 250 | 0.78 |
| 7_p ∥ | parallel | $Pb_{0.01}Bi_{0.49}Sb_{1.5}Te_3$ | 6.83 | 99.76 | $6.1 \times 10^{19}$ | 243 | |
| 7_p ⊥ | perpendicular | | | | $6.2 \times 10^{19}$ | 241 | 0.79 |

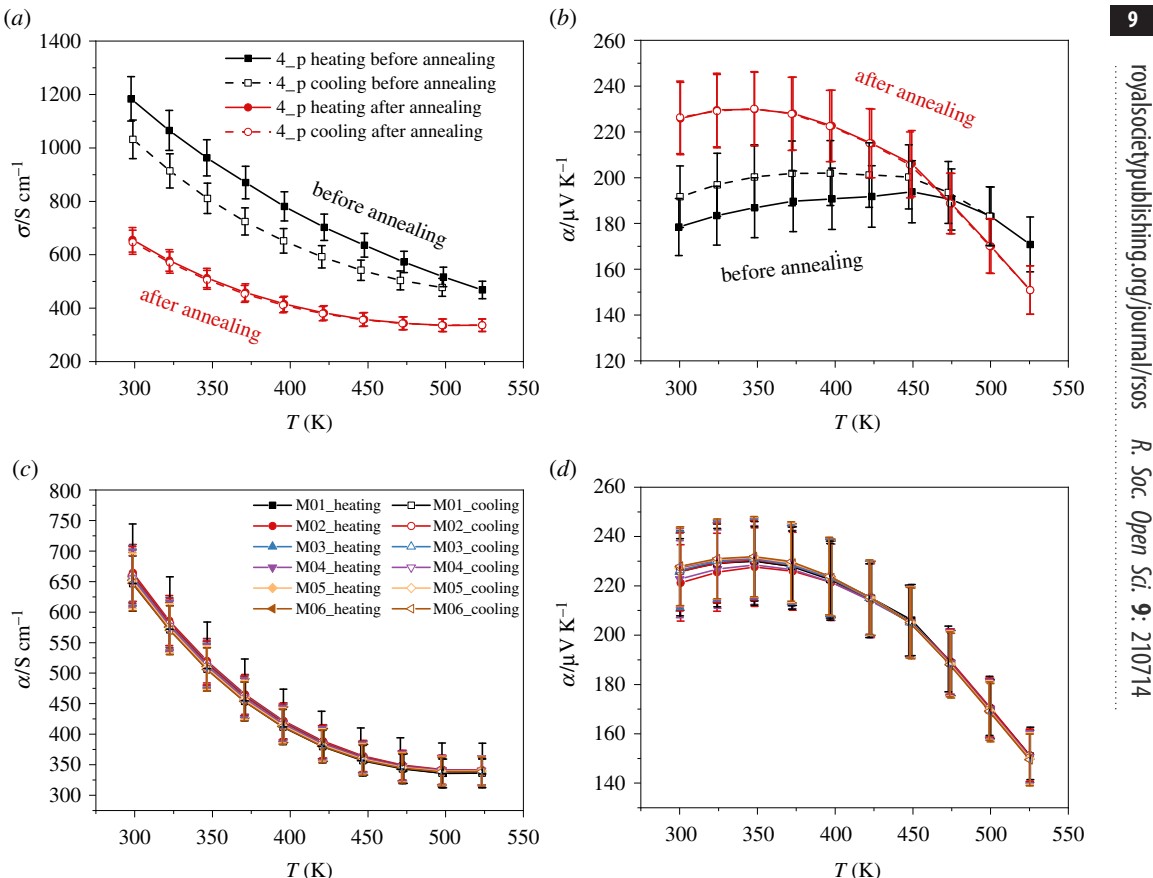

**Figure 4.** Influence of annealing on the electrical conductivity (*a*) and the Seebeck coefficient (*b*) and the long-term stability of the electrical conductivity (*c*) and the Seebeck coefficient (*d*) of the *p*-type sample 4_p. The sample was measured six times; the individual measurements were labelled M0x.

annealed sample 4_p ‖ was measured six times (figure 4*c,d*). There were no statistically significant differences in the measurement results.

## 3.4. Influence of texture on physical and thermoelectric data

As shown in table 3, the crystallites seem to be oriented randomly. To investigate the influence of the texture on the thermoelectric properties, they were measured perpendicular ⊥ and parallel ‖ to the pressing direction. Exemplary results of one n- and one p-type material are shown in figures 5 and 6. In the sample 1_n, the crystallites are significantly higher orientated in the a/b-plane perpendicular to the pressing direction (24.9%) than parallel to the pressing direction (13.3%). Consequently, the electrical and thermal conductivity is higher in the ⊥ direction. At higher temperatures, the difference for the electrical and thermal conductivity decreases from above 11% to 8% and 5%, respectively. Since the error bars overlap, the differences in the conductivities parallel and perpendicular to the pressing direction are not statistically significant. This result confirms the assumption of randomly oriented crystallites. In contrast with the conductivities, the Seebeck coefficient is isotropic [64–66], accordingly the Seebeck coefficients are very similar in both directions. Since both electrical and thermal conductivities are higher in the ⊥ direction, the *ZT* value is similar in both directions. The results are comparable to those reported by Yan *et al*. [66] and Lognoné *et al*. [67] for $Bi_2Te_{2.7}Se_{0.3}$ and $Bi_2Te_{2.4}Se_{0.6}$, respectively, which were synthesized in a similar way. However, the latter were able to increase the degree of texture by repeated sintering.

The same results can be found for the p-type sample 7_p. In this sample, however, the differences in the electrical conductivities in both directions are even smaller. This is due to the similar preferential orientation of the crystallites in the a/b plane perpendicular (18.9%) and parallel (22.1%) to the pressing direction. Nevertheless, the thermal conductivity differs by up to 10.8% depending on the orientation to the pressing direction. Still, the difference is not statistically significant for all measured thermoelectric properties.

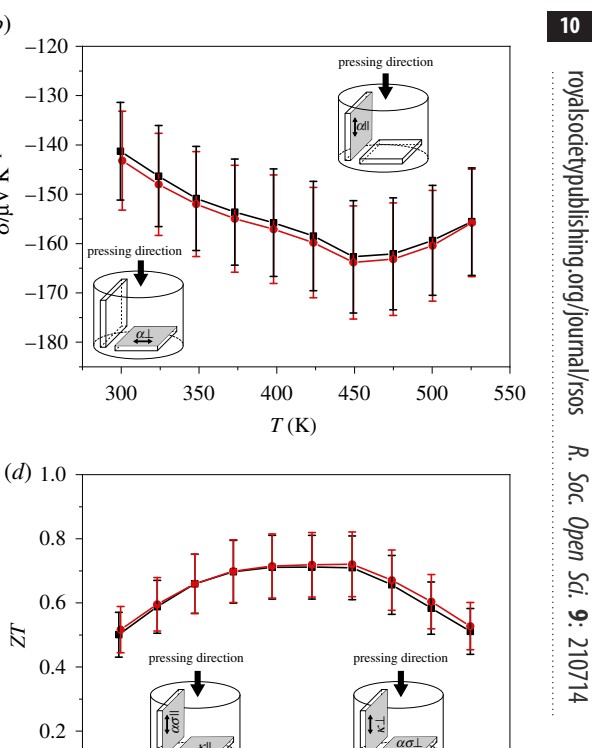

**Figure 5.** Electrical conductivity (*a*), Seebeck coefficient (*b*), thermal conductivity (*c*) and ZT (*d*) dependence of the temperature parallel and perpendicular to the pressing direction of 1_n Bi$_2$Te$_{2.7}$Se$_{0.3}$ + 0.1 wt% CHI$_3$.

## 3.5. Temperature-dependent properties

The thermoelectric properties as a function of the temperature of the n-type materials are shown in figure 7. As demonstrated previously, the physical transport properties of the materials are stable over the investigated temperature range. For this reason, only the data obtained during heating are shown.

The electrical conductivity decreases with the increasing temperature for all materials (figure 7*a*). This indicates that the n-type materials are degenerate semiconductors. The electrical conductivity values drop from above 900 S cm$^{-1}$ to below 655 S cm$^{-1}$. Although the carrier concentration of the sample 2_n is higher, the sample 1_n is a better electrical conductor in the entire temperature range, due to a pronounced change in charge carrier mobility. According to the relation $\sigma = e \cdot n \cdot \mu$, this leads to a higher electrical conductivity of the sample 1_n. The high carrier concentration of the sample 3_n leads to a low Seebeck coefficient (figure 7*b*). For all materials, the Seebeck coefficient increases with increasing temperature as expected. Above 450 K and 500 K for samples 1_n and 2_n, respectively, the Seebeck coefficient of these materials decreases, which corresponds to the bipolar effect [30,68]. This finding is expected for materials with a small band gap, like Bi$_2$Te$_3$, at high temperatures. The bipolar effect is based on the occurrence of the minority charge carriers (holes in n-type materials) in addition to the majority charge carriers (electrons in n-type materials) resulting in intrinsic conduction. Due to the increase of the energy band gap and carrier concentration with increasing Se content, the bipolar effect is suppressed and the maximum value for the Seebeck coefficient is shifted to higher temperatures [69]. Accordingly, the increased number of majority charge carrier concentration leads to significant changes in the temperature-dependent thermal conductivity as can be seen in figure 7*c*. The suppression of the bipolar effect for heavily substituted samples leads to a significantly lower thermal conductivity at elevated temperatures. Thus, the prominent increase in thermal conductivity at elevated temperature is not observed for the sample 3_n. [30,68] Furthermore, the lattice thermal conductivity decreases with a higher selenium content due to the mass-difference scattering (electronic supplementary material, figure S19) [3,70].

Overall, the suppression of the bipolar effect by increased Se content leads to a shift of the maximum ZT to higher temperatures, figure 7*d*. Between 400 and 450 K, the sample 1_n reaches the highest values

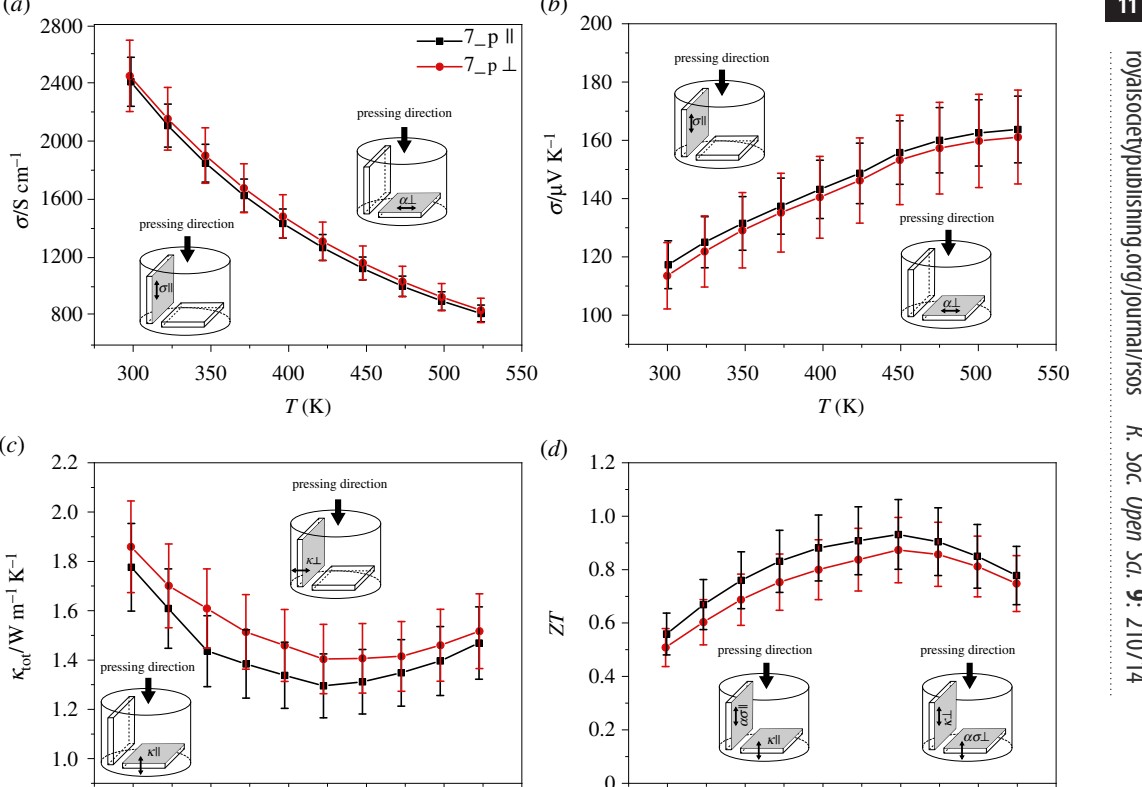

**Figure 6.** Electrical conductivity (*a*), Seebeck coefficient (*b*), thermal conductivity (*c*) and ZT (*d*) dependence of the temperature parallel and perpendicular to the pressing direction of 7_p $Pb_{0.01}Bi_{0.49}Sb_{1.5}Te_3$.

for *ZT* at around 0.71. The sample 2_n has its maximum *ZT* value between 450 and 500 K with around 0.67, while *ZT* of the sample 3_n increases with increasing temperature reaching 0.55 at *T* = 524 K. By using the best materials for the corresponding temperature range, the average *ZT* is 0.65 between 300 K and 525 K. The *ZT* values are in the same range as for $Bi_2Te_{3-y}Se_y$ [32,33] and for iodine-doped $Bi_2Te_3.Se_y$ [71] (*y* = 0.15–0.6), which were also produced by mechanochemical synthesis. However, for the synthesis of these materials, a five-times-longer milling time and a ball-to-powder ratio of 20 : 1 were used. Similar *ZT* values with 0.8 at 473 K were achieved by Pan *et al*. for $Bi_2Te_{2.2}Se_{0.8}$ after only 3 h of milling [72]. The comparisons with the results of other groups show that longer milling times have no further significant influence on the *ZT* value of the final product. Rather, thermoelectric properties can be significantly improved by subsequent preparation methods. E.g. Lognoné *et al*. were able to increase the *ZT* value of $Bi_2Te_{2.4}Se_{0.6}$ by 50% to almost 1 at 425 K by sintering the pellet a second time [67].

In figure 8, the thermoelectric properties of the p-type materials are shown as a function of the temperature. The substitution of Bi with Pb heavily increases the carrier concentration while decreasing the carrier mobilities to a lower extent (table 3). As a result, the electrical conductivities of the Pb-free samples 4_p and 5_p are below 700 S $cm^{-1}$, the values for the samples 6_p and 7_p are above 2400 S $cm^{-1}$ (figure 8*a*). Regardless of their chemical composition, all p-type materials exhibit degenerated semiconductor behaviour when heated. The significant increase in the carrier concentration by adding Pb leads to a reduction of the bipolar effect [27]. As a result, the Seebeck coefficient of Pb-containing materials increases (figure 8*b*), while it decreases for Pb-free materials with increasing temperature.

The high carrier concentrations of the samples 6_p and 7_p lead to almost twice as high thermal conductivity values at room temperature compared to samples without Pb (figure 8*c*). Due to the suppressed bipolar effect, the thermal conductivities of the samples 6_p and 7_p increase only slightly at high temperatures. In comparison, the thermal conductivities of samples 4_p and 5_p increase continuously with increasing temperature. Thus, the samples without Pb achieve higher thermal conductivities at high temperatures than the Pb-free samples.

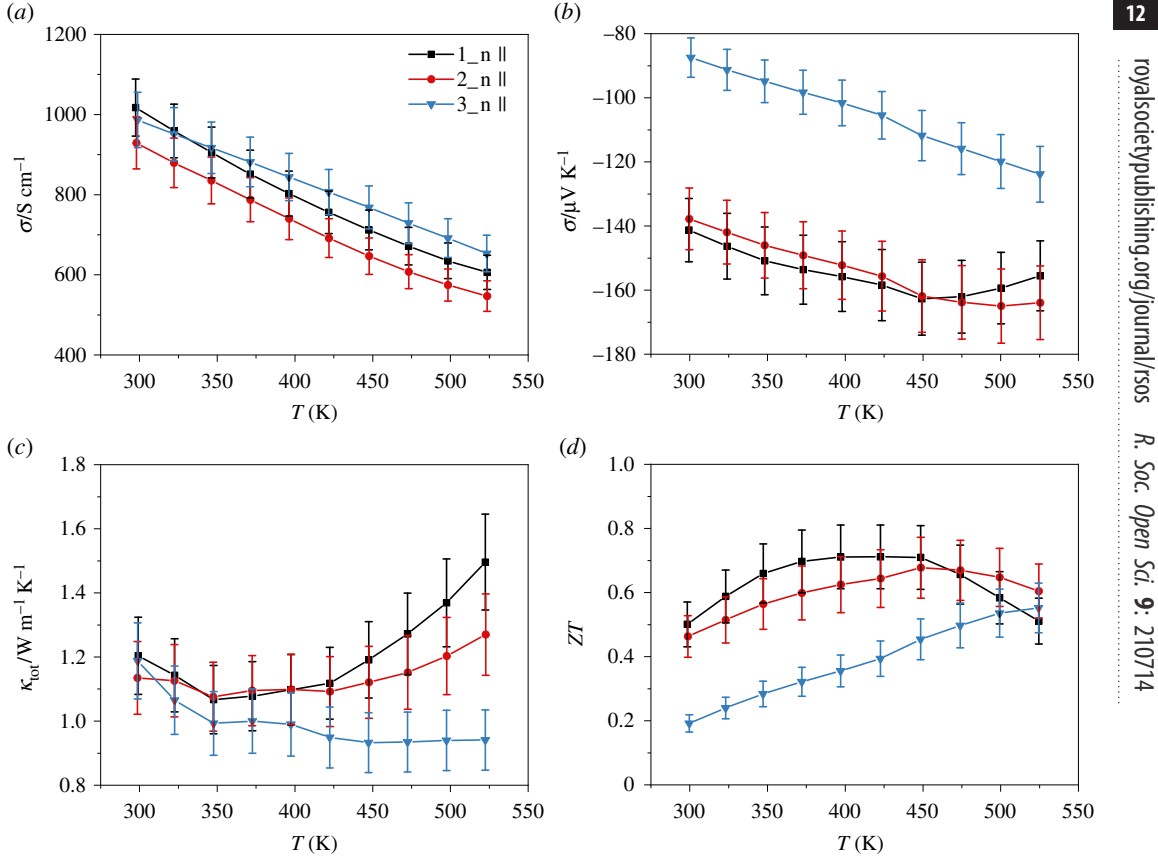

**Figure 7.** Temperature dependence of the electrical conductivity (*a*), Seebeck coefficient (*b*), thermal conductivity (*c*) and ZT (*d*) for the n-type materials $Bi_2Te_{2.7}Se_{0.3}$ + 0.1 wt% $CHI_3$ (1_n, black square), $Bi_2Te_{2.4}Se_{0.6}$ + 0.1 wt% $CHI_3$ (2_n, red circle) and $Bi_2Te_{1.5}Se_{1.5}$ 0.1 wt% $CHI_3$ (3_n, blue triangle).

As seen for the n-type materials, the maximum of *ZT* values shift to higher temperatures with increasing carrier concentration of the material, as expected. Around room temperature, the samples 4_p and 5_p reach the highest *ZT* = 0.90. For the sample 6_p, the highest *ZT* of 0.75 is observed around 474 K, while a slightly larger *ZT* of 0.93 is obtained for the sample at *T* = 449 K. With these materials, an average *ZT* value of 0.88 is achieved in the temperature range between 300 K and 550 K. The *ZT* values of the lead-free samples are in a similar range to those reported by Kitamura *et al.* [73] and Lee *et al.* [74] for $Bi_{0.3}Sb_{1.7}Te_3$ and $Bi_{0.5}Sb_{1.5}Te_3$, respectively. Although they have milled for 30 h and 4 h, respectively, there is no significant difference in the *ZT* values up to 400 K. The same can be observed with the comparison of the $Bi_{0.5}Sb_{1.5}Te_3$ with those prepared by Liu *et al.* [75]. Despite a longer milling time of 3 h, both the *ZT* values and the course are similar to that reported here. Comparing these results clarifies that by the combination of ball milling with FAST, no deterioration in the thermoelectric properties is observed, even with significantly shorter milling times.

## 4. Conclusion

We have shown that the described mechanochemical treatment of suitable starting samples can be used for a fast synthesis of phase-pure materials. Both n-type $Bi_2Se_xTe_{3-x}$ and p-type $Bi_{2-x}Sb_xTe_3$ were successfully prepared via MA and field-assisted sintering technology. It was proved that $Bi_2Te_3$ is formed after only 10 min and further milling has no influence on the phase composition. The XRD analysis and the results of the measurements of the physical properties depending on the orientation to the pressing direction prove that the crystallites are oriented randomly. By using 250 ml grinding jars, the presented method yielded 110 g of powders with each grinding, whereby scaling to larger quantities is possible. It is believed that, in particular, the MA process can be further optimized regarding the milling time by a detailed investigation of the milling parameters. The annealing process leads to stable thermoelectric properties up to 523 K. In addition, we analysed the influence of

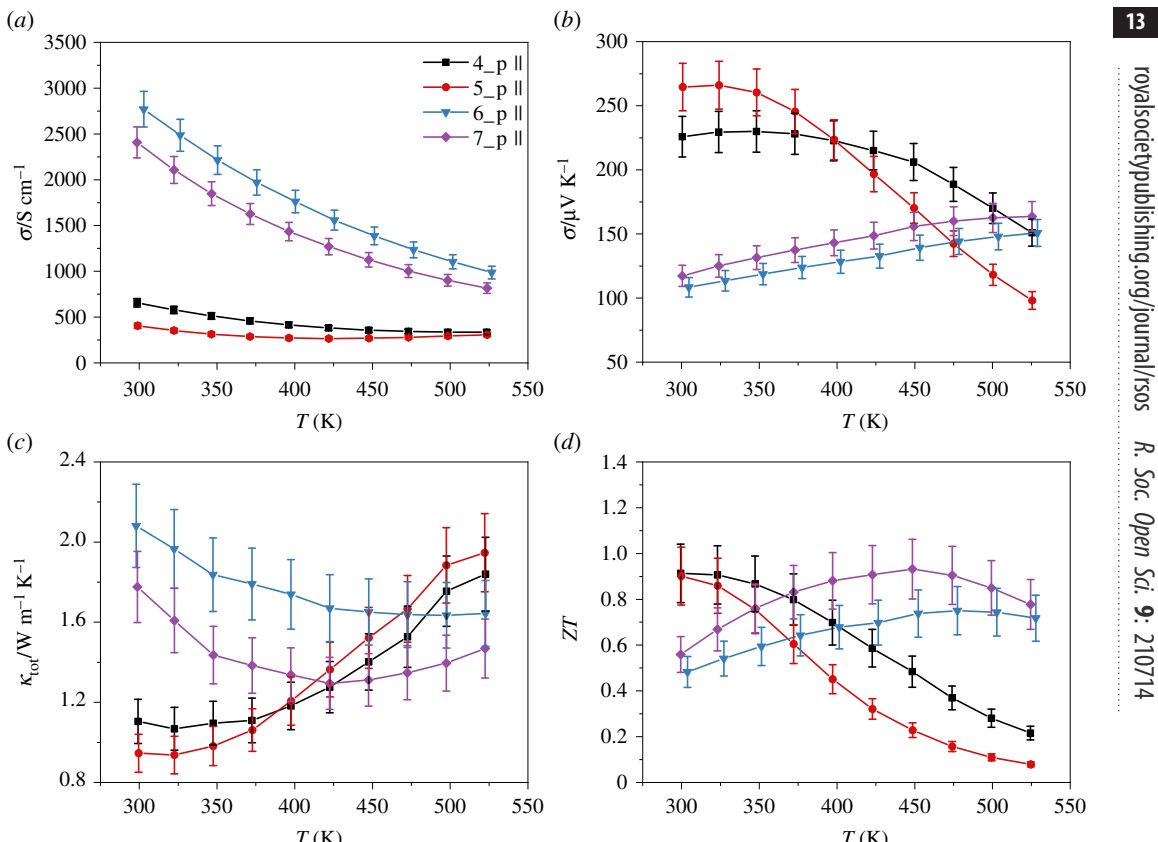

**Figure 8.** Temperature dependence of the electrical conductivity (*a*), Seebeck coefficient (*b*), thermal conductivity (*c*) and *ZT* (*d*) for the p-type materials $Bi_{0.3}Sb_{1.7}Te_3$ (4_p, black square), $Bi_{0.5}Sb_{1.5}Te_3$ (5_p, red circle), $Pb_{0.01}Bi_{0.29}Sb_{1.7}Te_3$ (6_p, blue triangle) and $Pb_{0.01}Bi_{0.49}Sb_{1.5}Te_3$ (7_p, purple diamond).

this method on the thermoelectric properties. In these attempts, no specific optimization of the material compositions and (nano)structuring regarding high *ZT* values was in the focus of these investigations. However, good *ZT* values were achieved despite the short milling time. The sample 1_n with the composition $Bi_2Te_{2.7}Se_{0.3} + 0.1$ wt% $CHI_3$ reached the highest *ZT* value with 0.71 of the n-type materials between 400 and 450 K. The p-type material (sample 6_p) with composition $Pb_{0.01}Bi_{0.29}Sb_{1.7}Te_3$ reached the highest *ZT* value of all samples with 0.93 at 449 K. By increasing the Se content in the n-type materials, the maximum of *ZT* was shifted to higher temperatures due to the suppression of the bipolar effect. In the p-type materials, this was accomplished by the substitution of Bi with Pb and increasing carrier concentration. In summary, the materials synthesized with these very short pre-reaction times behave like the well-known phase-pure, polycrystalline $(Bi,Sb)_2(Te,Se)_3$ compounds. This allows further optimization of the thermoelectric performance through nanostructuring while avoiding grain growth and nanoscaled precipitation due to these very short pre-reaction times. Thus, the described mechanochemical treatment allows not only further optimization of the materials, but also a fast synthesis of phase-pure and long-term stable materials on a large scale.

Ethics. Not relevant to this work.

Data accessibility. All data needed to evaluate the conclusions in the paper are present in the paper and/or the electronic supplementary material [76].

Authors' contributions. D.G.: conceptualization, data curation, formal analysis, funding acquisition, investigation, methodology, writing—original draft and writing—review and editing; J.D.K.: conceptualization, formal analysis, funding acquisition, project administration, supervision, validation, writing—original draft and writing—review and editing; M.P.: data curation, formal analysis, investigation, methodology, writing—original draft and writing—review and editing; H.G.: data curation, formal analysis, investigation, methodology, writing—original draft and writing—review and editing; W.B.: funding acquisition, project administration, supervision, validation, writing—original draft and writing—review and editing; L.K.: funding acquisition, project administration, supervision,

validation, writing—original draft and writing—review and editing; J.W.: funding acquisition, project administration, supervision, validation, writing—original draft and writing—review and editing.

All authors gave final approval for publication and agreed to be held accountable for the work performed therein.

Competing interests. We have no competing interests.

Funding. This work was supported by the German Research Foundation (DFG) within the research grant nos. BE1653/36-1, KI1263/16-1 and KO5397/2-1. The article processing charge was funded by the Baden-Wuerttemberg Ministry of Science, Research and Art and the University of Freiburg in the funding programme Open Access Publishing.

Acknowledgements. Not relevant to this work.

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
