## [Peer Review File · Royal Society Open Science]

Review History

RSOS-210714.R0 (Original submission)

Review form: Reviewer 1

Is the manuscript scientifically sound in its present form?

Yes

Are the interpretations and conclusions justified by the results?

Yes

Is the language acceptable?

Yes

Do you have any ethical concerns with this paper?

No

Have you any concerns about statistical analyses in this paper?

No

Recommendation?

Accept with minor revision (please list in comments)

Comments to the Author(s)

Dennies et al, have studied an influence of milling time on the phase formation of n- and p-type Bi₂Te₃ materials in the manuscript entitled "Time-dependent investigation of a mechanochemical synthesis of bismuth telluride-based materials and their structural and thermoelectric properties". Authors claim that Bi₂Te₃ forms just after 10minutes in a large scale and proved no further influence of phase composition upon milling. Further, authors have obtained the improved zTs of 0.71 and 0.93 for n- and p-type Bi₂Te₃, respectively. The manuscript is well written and the results were also clearly discussed. Thus, this manuscript can be accepted for publication in the reputed journal of Royal Society Open Science with the following corrections.

1. Authors should comment on the shift of Bragg peaks towards higher angle for one XRD pattern in Figure 2.
2. Discuss the reason behind notable (huge) reduction in the electrical conductivity of samples thus before annealing compared to after annealing in Figure 3 (a).
3. Authors should include the SEM micrographs and EDS results in the manuscript so as to view the microstructure and confirm the change in compositions, respectively.

Review form: Reviewer 2**Is the manuscript scientifically sound in its present form?**

Yes

Are the interpretations and conclusions justified by the results?

No

Is the language acceptable?

No

Do you have any ethical concerns with this paper?

No

Have you any concerns about statistical analyses in this paper?

No

Recommendation?

Reject

Comments to the Author(s)

In this manuscript, many variables were used in the experiments, which have not been deeply studied. Many experimental details are unknown. The content organization is relatively rough.

1. Bi₂Te₃ was formed by milling for only 10 minutes. What is the mechanism of the pre-reaction?
2. Page 4 and line 27, 'Before starting the measurement, the samples were coated with graphite to increase the emissivity and the absorbance.' What is the thickness of the graphite

coating? How to exclude the influence of graphite on thermal conductivity and thermoelectric performances?

3. Page 4 and line 51, "Each sample exhibits a distinct preferred orientation of 8-26 % of crystallites with a/b-plane in cutting direction (see Table 1)." How to calculate?
4. Page 4 and line 54, 'This preferred orientation seems to be random and not associated with uniaxial pressing direction or elemental constituents.' How to explain that the preferred orientations in samples 4p-7p have the same trend with the change of Bi and Sb content? Are there any repeated experiments for 1n-3n samples to prove that the preferred orientation is random rather than related to element content?
5. In Figure 3c, what are the M01-M06 curves?
6. Page 5 and line 43, ' Before annealing, the electrical conductivity appeared lower during cooling.' What is the essential reason for the phenomenon that the thermoelectric properties of the alloy are different during heating and cooling before annealing, while they are the same after annealing?
7. The description of the thermoelectric measurement system and the process should be described in more detail. For example, what is the separation between cool-end and hot-end? How is the temperature measured at the ends of alloy, etc.?

Review form: Reviewer 3

Is the manuscript scientifically sound in its present form?

No

Are the interpretations and conclusions justified by the results?

Yes

Is the language acceptable?

Yes

Do you have any ethical concerns with this paper?

Yes

Have you any concerns about statistical analyses in this paper?

No

Recommendation?

Major revision is needed (please make suggestions in comments)

Comments to the Author(s)

The main idea of study and goals needs more clear and better formulation, because the obtained results are far from the best ones obtained dozen years ago. Primary diffraction patterns are absent, only processed data are presented in fig. 1. It is difficult to understand presented results. In conclusion: This allows further optimization of the thermoelectric performance through nanostructuring while avoiding grain growth and nanoscaled precipitation. The presented results are interesting only in terms of the continuation of this work in the direction of nanostructured composites.

Review form: Reviewer 4

Is the manuscript scientifically sound in its present form?

Yes

Are the interpretations and conclusions justified by the results?

Yes

Is the language acceptable?

Yes

Do you have any ethical concerns with this paper?

Yes

Have you any concerns about statistical analyses in this paper?

No

Recommendation?

Major revision is needed (please make suggestions in comments)

Comments to the Author(s)

Submitted manuscript describes optimization of solid-state synthesis technique for n-type and p-type thermoelectric materials based on Bi₂Te₃. The authors find, that 10 minutes of mechanical milling with subsequent use of field-assisted sintering technique and annealing are sufficient to provide high-quality polycrystalline samples. The produced thermoelectric materials are characterized by low texture, which leads to almost isotropic transport properties resembling literature data, high mechanical durability, and good stability at elevated temperatures. The report constitutes improvement with respect to previous research on bismuth telluride, in most of which, mechanical milling was performed for several hours to achieve good thermoelectric performance.

Before considering the manuscript for publication, however, the authors need to clarify a major trouble regarding XRD phase analysis, results of which are gathered in Fig. 1 and Fig. 2. Going from 1 to 3 minutes of milling, amount of tellurium decreases from ca. 90 % to 70 % and Sb amount increases from ca. 10% to 27%. Bi content is overall very small, and authors show no Bi₂Te₃ in samples milled for 1 and 3 min. How elemental tellurium can be substituted by elemental antimony, if sample composition was the same for both milling times? As possible solution, the authors point out inhomogeneity of the studied samples. Do they suggest that if two XRD experiments were performed on different parts of the same milled sample, they could obtain different phase content? If this is the case, then results presented in Fig. 1. cannot be treated as conclusive. One of the most important messages of the manuscript is based on findings from XRD phases analysis; please address this important difficulty.

Furthermore, I would like to point out several minor issues:

1. Please include EDS chemical maps of the studied samples, which were used for preparation of Table 2. Why samples 3_n, 5_p, and 6_p were not examined?
2. Clarify the following fragment: "The Vickers hardness at room temperature of the in-plane samples are shown in Table 3. The pressing direction and within the uncertainty of the range. In hardness is around 1 GPa for the n-type materials, while the values of the p-type samples are lower in the range of 0.78 - 0.83."

3. "Furthermore, the lattice thermal conductivity decreases with a higher selenium content due to the mass-difference scattering (figure S12)" There is not figure S12 in the supplement, I guess the authors mentioned Figure S10.

Overall, the manuscript appears suitable for publication in Royal Society Open Science, if the troubles indicated in the review can be solved by the authors.

Decision letter (RSOS-210714.R0)

Dear Dr Groeneveld:

Title: Time-dependent investigation of a mechanochemical synthesis of bismuth telluride-based materials and their structural and thermoelectric properties

Manuscript ID: RSOS-210714

The editor assigned to your manuscript has now received comments from reviewers. We would like you to revise your paper in accordance with the referee and Subject Editor suggestions which can be found below (not including confidential reports to the Editor). Please note this decision does not guarantee eventual acceptance.

Please submit your revised paper before 18-Dec-2021. Please note that the revision deadline will expire at 00.00am on this date. If we do not hear from you within this time then it will be assumed that the paper has been withdrawn. In exceptional circumstances, extensions may be possible if agreed with the Editorial Office in advance. We do not allow multiple rounds of revision so we urge you to make every effort to fully address all of the comments at this stage. If deemed necessary by the Editors, your manuscript will be sent back to one or more of the original reviewers for assessment. If the original reviewers are not available we may invite new reviewers.

Please also include the following statements alongside the other end statements. As we cannot publish your manuscript without these end statements included, if you feel that a given heading is not relevant to your paper, please nevertheless include the heading and explicitly state that it is not relevant to your work.

- Ethics statement

Please clarify whether you received ethical approval from a local ethics committee to carry out your study. If so please include details of this, including the name of the committee that gave consent in a Research Ethics section after your main text. Please also clarify whether you received informed consent for the participants to participate in the study and state this in your Research Ethics section.

OR

Please clarify whether you obtained the necessary licences and approvals from your institutional animal ethics committee before conducting your research. Please provide details of these licences and approvals in an Animal Ethics section after your main text.

OR

Please clarify whether you obtained the appropriate permissions and licences to conduct the fieldwork detailed in your study. Please provide details of these in your methods section.

- Data accessibility

It is a condition of publication that you make available the data and research materials supporting the results in the article. Datasets should be deposited in an appropriate publicly available repository and details of the associated accession number, link or DOI to the datasets must be included in the Data Accessibility section of the article (<https://royalsocietypublishing.org/rsos/for-authors#question17>). Reference(s) to datasets should also be included in the reference list of the article with DOIs (where available).

Please include a Data Availability section after your main text stating where supporting data are available from, or where they will be made available should your article be accepted for publication.

If you wish to submit your supporting data or code to Dryad (<http://datadryad.org/>), or modify your current submission to dryad, please use the following link:
<http://datadryad.org/submit?journalID=RSOS&manu=RSOS-210714>

- Competing interests

Please include a Competing Interests section after your main text declaring any financial or non-financial competing interests. If you have no competing interests please state 'I/we have no competing interests.'

- Authors' contributions

Please include an Authors' Contributions section at the end of your main text detailing the contribution of each author. All authors should have read and approved the manuscript before submission and this should be stated in the Authors' Contributions section.

The list of Authors should meet all of the following criteria; 1) substantial contributions to conception and design, or acquisition of data, or analysis and interpretation of data; 2) drafting the article or revising it critically for important intellectual content; and 3) final approval of the version to be published.

AB carried out the molecular lab work, participated in data analysis, carried out sequence alignments, participated in the design of the study and drafted the manuscript; CD carried out the statistical analyses; EF collected field data; GH conceived of the study, designed the study,

coordinated the study and helped draft the manuscript. All authors gave final approval for publication.

• Acknowledgements

• Funding statement

Please include a funding section after your main text which lists the source of funding for each author.

Yours sincerely,
Dr Ellis Wilde
Publishing Editor, Journals

On behalf of the Subject Editor Professor Anthony Stace and the Associate Editor Dr Dattatray Late.

RSC Associate Editor
Comments to the Author:
Major revision is needed

RSC Subject Editor
Comments to the Author:
(There are no comments.)

Reviewers' Comments to Author:

Reviewer: 1

Comments to the Author(s)

Dennies et al, have studied an influence of milling time on the phase formation of n- and p-type Bi₂Te₃ materials in the manuscript entitled "Time-dependent investigation of a mechanochemical synthesis of bismuth telluride-based materials and their structural and thermoelectric properties". Authors claim that Bi₂Te₃ forms just after 10minutes in a large scale and proved no further influence of phase composition upon milling. Further, authors have obtained the improved zTs of 0.71 and 0.93 for n- and p-type Bi₂Te₃, respectively. The manuscript is well written and the results were also clearly discussed. Thus, this manuscript can be accepted for publication in the reputed journal of Royal Society Open Science with the following corrections.

1. Authors should comment on the shift of Bragg peaks towards higher angle for one XRD pattern in Figure 2.
2. Discuss the reason behind notable (huge) reduction in the electrical conductivity of samples thus before annealing compared to after annealing in Figure 3 (a).
3. Authors should include the SEM micrographs and EDS results in the manuscript so as to view the microstructure and confirm the change in compositions, respectively.

Reviewer: 2

Comments to the Author(s)

In this manuscript, many variables were used in the experiments, which have not been deeply studied. Many experimental details are unknown. The content organization is relatively rough.

1. Bi₂Te₃ was formed by milling for only 10 minutes. What is the mechanism of the pre-reaction?
2. Page 4 and line 27, 'Before starting the measurement, the samples were coated with graphite to increase the emissivity and the absorbance.' What is the thickness of the graphite coating? How to exclude the influence of graphite on thermal conductivity and thermoelectric performances?
3. Page 4 and line 51, "Each sample exhibits a distinct preferred orientation of 8-26 % of crystallites with a/b-plane in cutting direction (see Table 1)." How to calculate?
4. Page 4 and line 54, 'This preferred orientation seems to be random and not associated with uniaxial pressing direction or elemental constituents.' How to explain that the preferred orientations in samples 4p-7p have the same trend with the change of Bi and Sb content? Are there any repeated experiments for 1n-3n samples to prove that the preferred orientation is random rather than related to element content?
5. In Figure 3c, what are the M01-M06 curves?
6. Page 5 and line 43, 'Before annealing, the electrical conductivity appeared lower during cooling.' What is the essential reason for the phenomenon that the thermoelectric properties of the alloy are different during heating and cooling before annealing, while they are the same after annealing?
7. The description of the thermoelectric measurement system and the process should be described in more detail. For example, what is the separation between cool-end and hot-end? How is the temperature measured at the ends of alloy, etc.?

Reviewer: 3

Comments to the Author(s)

The main idea of study and goals needs more clear and better formulation, because the obtained results are far from the best ones obtained dozen years ago. Primary diffraction patterns are absent, only processed data are presented in fig. 1. It is difficult to understand presented results. In conclusion: This allows further optimization of the thermoelectric performance through nanostructuring while avoiding grain growth and nanoscaled precipitation. The presented results are interesting only in terms of the continuation of this work in the direction of nanostructured composites.

Reviewer: 4

Comments to the Author(s)

Submitted manuscript describes optimization of solid-state synthesis technique for n-type and p-type thermoelectric materials based on Bi₂Te₃. The authors find, that 10 minutes of mechanical milling with subsequent use of field-assisted sintering technique and annealing are sufficient to provide high-quality polycrystalline samples. The produced thermoelectric materials are characterized by low texture, which leads to almost isotropic transport properties resembling literature data, high mechanical durability, and good stability at elevated temperatures. The report constitutes improvement with respect to previous research on bismuth telluride, in most of

which, mechanical milling was performed for several hours to achieve good thermoelectric performance.

Before considering the manuscript for publication, however, the authors need to clarify a major trouble regarding XRD phase analysis, results of which are gathered in Fig. 1 and Fig. 2. Going from 1 to 3 minutes of milling, amount of tellurium decreases from ca. 90 % to 70 % and Sb amount increases from ca. 10% to 27%. Bi content is overall very small, and authors show no Bi₂Te₃ in samples milled for 1 and 3 min. How elemental tellurium can be substituted by elemental antimony, if sample composition was the same for both milling times? As possible solution, the authors point out inhomogeneity of the studied samples. Do they suggest that if two XRD experiments were performed on different parts of the same milled sample, they could obtain different phase content? If this is the case, then results presented in Fig. 1. cannot be treated as conclusive. One of the most important messages of the manuscript is based on findings from XRD phases analysis; please address this important difficulty.

Furthermore, I would like to point out several minor issues:

1. Please include EDS chemical maps of the studied samples, which were used for preparation of Table 2. Why samples 3_n, 5_p, and 6_p were not examined?
2. Clarify the following fragment: "The Vickers hardness at room temperature of the in-plane samples are shown in Table 3. The pressing direction and within the uncertainty of the range. In hardness is around 1 GPa for the n-type materials, while the values of the p-type samples are lower in the range of 0.78 - 0.83."
3. "Furthermore, the lattice thermal conductivity decreases with a higher selenium content due to the mass-difference scattering (figure S12)" There is not figure S12 in the supplement, I guess the authors mentioned Figure S10.

Overall, the manuscript appears suitable for publication in Royal Society Open Science, if the troubles indicated in the review can be solved by the authors.

Author's Response to Decision Letter for (RSOS-210714.R0)

See Appendix A.

RSOS-210714.R1 (Revision)

Review form: Reviewer 3

Is the manuscript scientifically sound in its present form?

Yes

Are the interpretations and conclusions justified by the results?

Yes

Is the language acceptable?

Yes

Do you have any ethical concerns with this paper?

No

Have you any concerns about statistical analyses in this paper?

No

Recommendation?

Accept with minor revision (please list in comments)

Comments to the Author(s)

The authors have every opportunity to reach the limiting level of ZT in this class of materials, including phonon design as a tool. However, the purpose of this particular manuscript needs to be clarified.

Review form: Reviewer 4

Is the manuscript scientifically sound in its present form?

Yes

Are the interpretations and conclusions justified by the results?

Yes

Is the language acceptable?

Yes

Do you have any ethical concerns with this paper?

No

Have you any concerns about statistical analyses in this paper?

No

Recommendation?

Accept as is

Comments to the Author(s)

The authors made appropriate changes to the manuscript. I recommend it for publication in the current form.

Decision letter (RSOS-210714.R1)

Dear Dr Groeneveld:

Title: Time-dependent investigation of a mechanochemical synthesis of bismuth telluride-based materials and their structural and thermoelectric properties

Manuscript ID: RSOS-210714.R1

Thank you for submitting the above manuscript to Royal Society Open Science. On behalf of the Editors and the Royal Society of Chemistry, I am pleased to inform you that your manuscript will be accepted for publication in Royal Society Open Science subject to minor revision in accordance with the referee suggestions. Please find the reviewers' comments at the end of this email.

The reviewers and handling editors have recommended publication, but also suggest some minor revisions to your manuscript. Therefore, I invite you to respond to the comments and revise your manuscript.

Please also include the following statements alongside the other end statements. As we cannot publish your manuscript without these end statements included, if you feel that a given heading is not relevant to your paper, please nevertheless include the heading and explicitly state that it is not relevant to your work. We have included a screenshot example of the end statements for reference.

- Ethics statement

Please clarify whether you received ethical approval from a local ethics committee to carry out your study. If so please include details of this, including the name of the committee that gave consent in a Research Ethics section after your main text. Please also clarify whether you received informed consent for the participants to participate in the study and state this in your Research Ethics section.

OR

Please clarify whether you obtained the necessary licences and approvals from your institutional animal ethics committee before conducting your research. Please provide details of these licences and approvals in an Animal Ethics section after your main text.

OR

Please clarify whether you obtained the appropriate permissions and licences to conduct the fieldwork detailed in your study. Please provide details of these in your methods section.

- Data accessibility

It is a condition of publication that you make available the data and research materials supporting the results in the article. Datasets should be deposited in an appropriate publicly available repository and details of the associated accession number, link or DOI to the datasets must be included in the Data Accessibility section of the article (<https://royalsocietypublishing.org/rsos/for-authors#question17>). Reference(s) to datasets should also be included in the reference list of the article with DOIs (where available).

Please include a Data Availability section after your main text stating where supporting data are available from, or where they will be made available should your article be accepted for publication.

<http://datadryad.org/submit?journalID=RSOS&manu=RSOS-210714.R1>

- Competing interests

Please include a Competing Interests section after your main text declaring any financial or non-financial competing interests. If you have no competing interests please state 'I/we have no competing interests.'

- Authors' contributions

Please include an Authors' Contributions section at the end of your main text detailing the contribution of each author. All authors should have read and approved the manuscript before submission and this should be stated in the Authors' Contributions section.

The list of Authors should meet all of the following criteria; 1) substantial contributions to conception and design, or acquisition of data, or analysis and interpretation of data; 2) drafting the article or revising it critically for important intellectual content; and 3) final approval of the version to be published.

- Acknowledgements

- Funding statement

Please include a funding section after your main text which lists the source of funding for each author.

Because the schedule for publication is very tight, it is a condition of publication that you submit the revised version of your manuscript before 19-Feb-2022. Please note that the revision deadline will expire at 00.00am on this date. If you do not think you will be able to meet this date please let me know immediately.

- 1) A text file of the manuscript (tex, txt, rtf, docx or doc), references, tables (including captions) and figure captions. Do not upload a PDF as your "Main Document".
- 2) A separate electronic file of each figure (EPS or print-quality PDF preferred (either format should be produced directly from original creation package), or original software format)
- 3) Included a 100 word media summary of your paper when requested at submission. Please ensure you have entered correct contact details (email, institution and telephone) in your user account

- 4) Included the raw data to support the claims made in your paper. You can either include your data as electronic supplementary material or upload to a repository and include the relevant doi within your manuscript
- 5) All supplementary materials accompanying an accepted article will be treated as in their final form. Note that the Royal Society will neither edit nor typeset supplementary material and it will be hosted as provided. Please ensure that the supplementary material includes the paper details where possible (authors, article title, journal name).

Kind regards,
Dr Ellis Wilde
Publishing Editor, Journals

On behalf of the Subject Editor Professor Anthony Stace and the Associate Editor Dr Dattatray Late.

RSC Associate Editor
Comments to the Author:
Accept with minor revisions

RSC Subject Editor
Comments to the Author:
(There are no comments.)

Reviewer comments to Author:
Reviewer: 3

Comments to the Author(s)
The authors have every opportunity to reach the limiting level of ZT in this class of materials, including phonon design as a tool. However, the purpose of this particular manuscript needs to be clarified.

Reviewer: 4

Comments to the Author(s)

The authors made appropriate changes to the manuscript. I recommend it for publication in the current form.

Author's Response to Decision Letter for (RSOS-210714.R1)

See Appendix B.

Decision letter (RSOS-210714.R2)

Dear Dr Groeneveld:

Title: Time-dependent investigation of a mechanochemical synthesis of bismuth telluride-based materials and their structural and thermoelectric properties

Manuscript ID: RSOS-210714.R2

It is a pleasure to accept your manuscript in its current form for publication in Royal Society Open Science. The chemistry content of Royal Society Open Science is published in collaboration with the Royal Society of Chemistry.

Yours sincerely,

Kate Jones

Publishing Editor, Journals

On behalf of the Subject Editor Professor Anthony Stace and the Associate Editor Dr Dattatray Late.

RSC Associate Editor
Comments to the Author:
Accept as is

Reviewer(s)' Comments to Author:

Appendix A

Dear Dr. Wilde,

we are pleased about the constructive feedback from the reviewers and have revised our manuscript in accordance with their comments and suggestions.

Our answers to the comments of the reviewers are marked in **blue**, while changes in the manuscript are marked in **green**.

We thank you and the reviewers for carefully reading and commenting the manuscript and are grateful for the advice.

Best regards,

Dennis Groeneveld

Reviewer #1 - Comments to the Author

Bi_2Te_3 materials in the manuscript entitled “Time-dependent investigation of a mechanochemical synthesis of bismuth telluride-based materials and their structural and thermoelectric properties”. Authors claim that Bi_2Te_3 forms just after 10 minutes in a large scale and proved no further influence of phase composition upon milling. Further, authors have obtained the improved zTs of 0.71 and 0.93 for n- and p-type Bi_2Te_3 , respectively. The manuscript is well written and the results were also clearly discussed. Thus, this manuscript can be accepted for publication in the reputed journal of Royal Society Open Science with the following corrections.

We thank the reviewer for the positive words.

1. Authors should comment on the shift of Bragg peaks towards higher angle for one XRD pattern in Figure 2.

We appreciate the comment. The samples with the biggest shift to higher angles are the samples 3, 4 and 6. These samples have either high content of Se (sample 3) or high content of Sb (sample 4 and 6). So in comparison to the other samples the diffraction angles are observed at higher values, because the mean atomic radii are smaller and therefore the unit cells are also smaller.

2. Discuss the reason behind notable (huge) reduction in the electrical conductivity of samples thus before annealing compared to after annealing in Figure 3 (a).

During the annealing, materials are heated below their melting temperature for a longer period of time. In this way, defects within the material are healed by the diffusion of the atoms. As a result, the charge carrier mobility increases, while the carrier concentration decreases to a greater extent. Thus, the overall electrical conductivity decreases and the Seebeck coefficient increases after annealing. To clarify the results, the carrier concentration and mobility before annealing were added to the Electronic Supplementary Information and the following sentences have been added to p. 4, lines 44-49 in the manuscript:

Due to the annealing the charge carrier mobility increases, while the charge carrier concentration decreases to a greater extent (see table 3 and table S3). Therefore, the overall electrical conductivity decreases and the Seebeck coefficient increases after annealing.

Table S3. Overview of the carrier concentration and mobility at room temperature for the synthesized thermoelectric materials before annealing

Sample	Chemical formula	Carrier concentration cm^{-3}	Carrier mobility $\text{cm}^2 \text{V}^{-1} \text{s}^{-1}$
1_n	$\text{Bi}_2\text{Te}_{2.7}\text{Se}_{0.3}$ + 0.1 wt% CHI_3	$4.5 \cdot 10^{19}$	151
2_n	$\text{Bi}_2\text{Te}_{2.4}\text{Se}_{0.6}$ + 0.1 wt% CHI_3	$5.5 \cdot 10^{19}$	111
3_n	$\text{Bi}_2\text{Te}_{1.5}\text{Se}_{1.5}$ + 0.1 wt% CHI_3	$6.8 \cdot 10^{19}$	94
4_p	$\text{Bi}_{0.3}\text{Sb}_{1.7}\text{Te}_3$	$2.9 \cdot 10^{19}$	253
5_p	$\text{Bi}_{0.5}\text{Sb}_{1.5}\text{Te}_3$	$1.2 \cdot 10^{19}$	265
6_p	$\text{Pb}_{0.01}\text{Bi}_{0.29}\text{Sb}_{1.7}\text{Te}_3$	$8.6 \cdot 10^{19}$	206
7_p	$\text{Pb}_{0.01}\text{Bi}_{0.49}\text{Sb}_{1.5}\text{Te}_3$	$7.5 \cdot 10^{19}$	202

3. Authors should include the SEM micrographs and EDS results in the manuscript so as to view the microstructure and confirm the change in compositions, respectively.

Thank you. SEM surface micrographs of samples 1_n, 2_n, 4_p and 7_p have been added to the manuscript as figure 3, with the text in chapter 4.1, p. 4, lines 12-15 being changed to address and reference the new information. Concerning the EDS data it is not clear what changes are meant.

Figure 1. SEM surface micrographs of samples 1_n, 2_n, 4_p and 7_p before annealing, imaged at identical magnifications. All samples exhibit slightly grained surfaces, with the particle size increasing significantly from n-type to p-type materials.

The samples' surfaces exhibited a visible grain structure, with particle sizes ranging from 1-2 μm for both n-type Bi-Te-Se materials and 2-4 μm for the p-type Bi-Te-Sb systems. Before annealing, all investigated samples appeared to be homogeneous, both in regard to structure and composition (see Figure 1 and supporting information figure S15-S18). For the n-type materials, no significant changes could be observed after annealing.

Reviewer #2 - Comments to the Author

In this manuscript, many variables were used in the experiments, which have not been deeply studied. Many experimental details are unknown. The content organization is relatively rough. We thank the reviewer for carefully reading our manuscript and the valuable comments.

1. Bi_2Te_3 was formed by milling for only 10 minutes. What is the mechanism of the pre-reaction? During milling the reactants are mixed and ground into a fine powder. By the collision of the milling balls with the powder a type of welding occurs. Further milling results in repeated cold welding and re-breaking of the powder particles. The XRD analysis revealed the initial formation of Bi_2Te_3 , presumably because it is thermodynamically more stable than Sb_2Te_3 .

2. Page 4 and line 27, 'Before starting the measurement, the samples were coated with graphite to increase the emissivity and the absorbance.' What is the thickness of the graphite coating? How to exclude the influence of graphite on thermal conductivity and thermoelectric performances? Thank you for the comment. The graphite was applied as a graphite spray and is very thin (generally a few micrometers) compared to the sample, which has a thickness of about 1 mm. Therefore, the influence of the graphite layer on the thermal diffusivity is negligible.

3. Page 4 and line 51, "Each sample exhibits a distinct preferred orientation of 8-26 % of crystallites with a/b-plane in cutting direction (see Table 1)." How to calculate?

The way how we obtained this values is mentioned in the experimental section in the part related to X-ray diffraction. We have done it by comparing the observed X-ray diffraction intensities of the pelletized samples with those of the grinded samples. In this way we compared the textured sample with the same but untextured sample. Using the well known approach of March and Dollase we calculated the intensity ratios of the 006 and 2-10 reflections which can be converted in a percentual expression of preferred orientation.

4. Page 4 and line 54, 'This preferred orientation seems to be random and not associated with uniaxial pressing direction or elemental constituents.' How to explain that the preferred orientations in samples 4p-7p have the same trend with the change of Bi and Sb content? Are there any repeated experiments for 1n-3n samples to prove that the preferred orientation is random rather than related to element content?

Sample 4, 6 and 7 show the same trend, but sample 5 shows opposite trend. The low number of data sets allowed us not to look for a possible trend. Two points always show a trend.

5. In Figure 3c, what are the M01-M06 curves?

Sample 4_p was measured six times at the SBA. M0x indicates the individual measurements.

Figure 4. Influence of annealing on the electrical conductivity (a) and the Seebeck coefficient (b) and the long-term stability of the electrical conductivity (c) and the Seebeck coefficient (d) of the p-type sample 4_p. The sample was measured six times, the individual measurements were labeled M0x.

6. Page 5 and line 43, ' Before annealing, the electrical conductivity appeared lower during cooling.' What is the essential reason for the phenomenon that the thermoelectric properties of the alloy are different during heating and cooling before annealing, while they are the same after annealing? Heating during the measurement leads to the healing of defects. As a result, the charge carrier mobility is increased, but at the same time the charge carrier concentration is reduced. If, however, the samples had already been annealed beforehand, a large proportion of the defects were healed and the material is in a thermodynamically stable state. Thus, there is no change in the physical properties even with further temperature treatment.

7. The description of the thermoelectric measurement system and the process should be described in more detail. For example, what is the separation between cool-end and hot-end? How is the temperature measured at the ends of alloy, etc.?

The electrical conductivity and the Seebeck coefficient were measured using the SBA 458 Nemesis[®] and the thermal diffusivity was measured with the LFA 457 MicroFlash[®] apparatus, both from Netzsch. For the measurement of the electrical conductivity, an electrical current I is induced several times in both directions into the sample via two current pins. The voltage is measured via two thermocouples on the sample's lower surface. By adding the sample thickness, the resistivity or electrical conductivity is then calculated. Two micro heaters are placed below the two ends of the sample, generating temperature gradients in both directions. Both the temperature and the resulting Seebeck voltage are measured via the thermocouples and then used to calculate the Seebeck coefficient.

For the measurement of thermal diffusivity, the lower surface of the sample is heated by a short laser pulse. Above the sample is an infrared detector, which measures the resulting temperature change on the upper surface. The thermal diffusivity can then be calculated from the signal height and the sample thickness. The specific heat capacity of the sample can be determined by measuring a reference material with a known specific heat capacity. Together with the density of the sample, the thermal conductivity can be determined, neglecting the temperature dependence of the density.

Reviewer #3 - Comments to the Author

The main idea of study and goals needs more clear and better formulation, because the obtained results are far from the best ones obtained dozen years ago. Primary diffraction patterns are absent, only processed data are presented in fig. 1. It is difficult to understand presented results. In conclusion: This allows further optimization of the thermoelectric performance through nanostructuring while avoiding grain growth and nanoscaled precipitation. The presented results are interesting only in terms of the continuation of this work in the direction of nanostructured composites.

We thank the reviewer for thoroughly reading our manuscript. Primary diffraction data is available in the electronic supplementary information.

Reviewer #4 - Comments to the Author

Submitted manuscript describes optimization of solid-state synthesis technique for n-type and p-type thermoelectric materials based on Bi_2Te_3 . The authors find, that 10 minutes of mechanical milling with subsequent use of field-assisted sintering technique and annealing are sufficient to provide high-quality polycrystalline samples. The produced thermoelectric materials are characterized by low texture, which leads to almost isotropic transport properties resembling literature data, high mechanical durability, and good stability at elevated temperatures. The report constitutes improvement with respect to previous research on bismuth telluride, in most of which, mechanical milling was performed for several hours to achieve good thermoelectric performance.

We appreciate the valuable comments and thank the reviewer for thoroughly reading our manuscript.

Before considering the manuscript for publication, however, the authors need to clarify a major trouble regarding XRD phase analysis, results of which are gathered in Fig. 1 and Fig. 2. Going from 1 to 3 minutes of milling, amount of tellurium decreases from ca. 90 % to 70 % and Sb amount increases from ca. 10% to 27%. Bi content is overall very small, and authors show no Bi_2Te_3 in samples milled for 1 and 3 min. How elemental tellurium can be substituted by elemental antimony, if sample composition was the same for both milling times? As possible solution, the authors point out inhomogeneity of the studied samples. Do they suggest that if two XRD experiments were performed on different parts of the same milled sample, they could obtain different phase content? If this is the case, then results presented in Fig. 1. cannot be treated as conclusive. One of the most important messages of the manuscript is based on findings from XRD phases analysis; please address this important difficulty.

This is an important point. In the beginning of the milling process the material is inhomogeneous because not all components reacted already to form the final samples. After <5 min the observed elemental composition is not identical at different spots in the same sample. For a homogeneous distribution >10 minutes of milling are needed. To be on the safe side the milling process was still done for 45 minutes before FAST treatment. The samples which are later characterized and discussed in the manuscript are homogeneous in composition.

1. Please include EDS chemical maps of the studied samples, which were used for preparation of Table 2. Why samples 3_n, 5_p, and 6_p were not examined?

The reviewer's request is appreciated. EDS chemical maps of samples 1_n, 2_n, 4_p and 7_p have been measured and were added to the electronic supplementary information as figures S15-S18.

They are further referenced on p. 4, lines 13-15 as:

Before annealing, all investigated samples appeared to be homogeneous, both in regard to structure and composition (see Figure 1 and supporting information figure S15-S18).

Samples 3_n, 5_p and 6_P could not be analyzed as the SEM/EDX was out of order at this time.

2. Clarify the following fragment: “The Vickers hardness at room temperature of the in-plane samples are shown in Table 3. The pressing direction and within the uncertainty of the range. In hardness is around 1 GPa for the n-type materials, while the values of the p-type samples are lower in the range of 0.78 – 0.83.”

We thank the reviewer for this comment. There is no simple relationship between the hardness of a material and other properties like bond strength, covalency, strain and strength parameters, dimensionality of the structure etc. In our case we observe a relationship between the density and hardness of the samples: the density of the p-type materials is lower than that of the n-type samples. This correlates with the hardness data.

The last two sentences in p. 4, lines 37-39 have been changed to make the statement clearer.

The Vickers hardness at room temperature of the in-plane samples are shown in Table 3. In-plane hardness could be determined as around 1 GPa for the n-type materials, while the hardness of the p-type-samples appears lower, in the range of 0.78-0.83 GPa.”

3. “Furthermore, the lattice thermal conductivity decreases with a higher selenium content due to the mass-difference scattering (figure S12)” There is not figure S12 in the supplement, I guess the authors mentioned Figure S10.

Thank you. This is correct and is has been changed in p. 5, lines 38-39.

Furthermore, the lattice thermal conductivity decreases with a higher selenium content due to the mass-difference scattering (figure S19) [3,70].

Appendix B

Dear Dr. Wilde,

thank you for accepting our manuscript for publication. We are pleased about the constructive feedback from the reviewers and have revised our manuscript in accordance with their comments. Our answers to the comments of the reviewers are marked in **blue**, while changes in the manuscript are marked in **green**.

We thank you and the reviewers for carefully reading and commenting the manuscript and are grateful for the advice.

Best regards,

Dennis Groeneveld

Reviewer #3 - Comments to the AuthorThe authors have every opportunity to reach the limiting level of ZT in this class of materials, including phonon design as a tool. However, the purpose of this particular manuscript needs to be clarified.

Thank you for your comment. In our manuscript, we investigate in detail a fast method for the preparation of phase-pure thermoelectric bismuth tellurides in large quantities. In addition, we show further optimization potential enabled by this method for further improvement of the materials.

To further emphasize the purpose of this manuscript, a sentence was added in the conclusion on p. 6, lines 40-41:

Thus, the described mechanochemical treatment allows not only further optimization of the materials, but also a fast synthesis of phase-pure and long-term stable materials on a large scale.

Reviewer #4 - Comments to the Author

The authors made appropriate changes to the manuscript. I recommend it for publication in the current form.

Thank you very much.